

# Science, Poetry, and Music for Landscapes of the Marche Region, Italy. Teaching the Conservation of Natural Heritage

Olivia Nesci[1]and Laura Valentini[2]

[1] Department of Pure and Applied Sciences, University of Urbino Carlo Bo, Urbino (PU), 61029, Italy
[2] Department of Biomolecular Sciences, University of Urbino Carlo Bo, Urbino (PU), 61029, Italy

*Correspondence to*: Laura Valentini (laura.valentini@uniurb.it)

**Abstract.** We present a new approach in science communication that uses artistic works to entice people to learn about landscapes. To this aim, we use narratives about a place in plain language  accompanied by  visual stimulations, poetry, and ancient music. The multidisciplinary approach resulting from the encounter and interplay among the different communicative
methods arouse an emotional and intellectual experience that enables a personal connection to the place. This work is part of a larger multidisciplinary project covering 20 sites in the Marche Region (Central Italy) that includes scientific information on geological-geomorphological genesis, trekking itineraries, poetry, ancient music, video and cultural offerings. The project is documented through live multidisciplinary performances, the publication of project materials in a book with a DVD attached, and through a web site with the same contents. Among the many amazing landscapes of the Marche Region, we focus here on
three sites from the north, the centre and the south of the region:  "The sea-cliff of San Bartolo", "The flatiron of Mount Petrano" and "The fault of Mount Vettore", chosen as examples for their different processes of genesis and evolution. Our goal is to promote a deeper understanding of landscapes by integrating their origin and physical aesthetic with their cultural and artistic heritage. In doing so, we intend to educate people to have a new perception of geo-sites, starting from its physical beauty, building on scientific study and cultural history, and arriving to the knowledge of its social importance.

**1 Introduction**

Ordinary people are often turned off by scientific language which can seem foreign and dry despite its importance. In contrast, the arts have often been used in history to successfully communicate, influence, and educate. Art is, in fact, able to motivate a large audience and art works are an important component in several contexts: protests or environmental movements promoting important social issues, and for raising awareness of social and environmental problems linked to scientific contents (Jordan,
2008; Curtis, 2011; 2012). The arts can help synthesize and convey complex scientific information, promote new ways of looking at issues, touch people's emotions, and create a celebratory and positive atmosphere (Curtis et al., 2012). The arts have nowadays been rediscovered as an effective medium for conveying science to the public. Tracking used approaches may ultimately help to establish ways to achieve more impactful outcomes, and to measure the effectiveness of arts-based science communication for raising awareness people to complex topics. (Lesen et al., 2016). Evidence suggests that the arts can deeply



engage people by focusing on the effective domain of learning (i.e. engagement, attitude, emotion) rather than on the cognitive domain (i.e. comprehension, application), which is often emphasized in science education (Friedman, 2013). Some affirm that, by utilizing both domains, arts-based science communication catalyses attention and creativity by encouraging intuitive thinking (Scheffer et al., 2015). Our work, in harmony and agreement with such research, combines scientific communication with arts (poetry and music), aiming to expose the public to the great richness of information and beauty represented by a

landscape, and at the same time,  provides evidence of a methodology that can attract visitors to learn and experience such places.

The geological literature linked to geotourism, cultural heritage, and aesthetics, has been very numerous in the last twenty years (Reynard et al., 2007; Coratza and Panizza, 2009; Gordon, 2018 and reference therein). Recently, newer ideas emerged about the physical landscape, not directly linked to geo-tourism but, as a way of transmitting science to the public in order to

increase curiosity and passion that then could also drive tourism (Lanza and Negrete, 2007). The definition given by the European Landscape Convention (The landscape is part of the land, as perceived by local people or visitors, which evolves through time as a result of being acted upon by natural forces and human beings, Council of Europe, 2000) best expresses the meaning of landscape. In fact, the concept of perception by the human community is held in high regard. But perception is different to the extent that attention is high or low, and this depends on the emotional state of the beholder and the interest he

or she has for that place. Interest arises from the curiosity to know how and why that place was formed: in practice, its geological history.

The Marche is an interesting region from many points of view. The region is still unspoiled in its multiple landscape identities: wild promontories on the sea and delightful sandy beaches, rugged mountain landscapes and soft green hills, spectacular karst caves, deep gorges. This wealth is a consequence of the geological history of the Apennine chain, which produced amazing

contrasts of physical forms in a very limited space. These places have been, since the dawn of history, sites of very important human settlements, leaving us several testimonies of great cultural interest. Around these many beautiful landscapes, it is not difficult to find places of incomparable value: towns and villages full of charm, important historic communication routes, extraordinary cultural, artistic, and architectural riches, religious places, and popular local traditions. The region has all the requirements to make the natural environment its strong point for driving public interest, namely enormous biodiversity based

upon microclimatic, floristic, and faunal variety. The Marche region also saw the birth of great personalities, while many others have travelled through these lands. A geologist who observes a landscape does not just grasp its beauty but starts a mental process that refers back to the complex mechanisms that generated and shaped it, only then to perceive its critical issues and environmental fragility. This understanding inevitably produces a personal appreciation and attachment to the place, even a love that would not otherwise be born.

Nevertheless, people commonly think of the scientific approach and language as too technical, or foreign, and therefore they often instinctively reject it or fail to understand its message. Geoscientists should be encouraged to transmit their technical expertise more effectively to a non-technical audience. Some forms of communication related to art, such as poetry and music, directly addresses the emotional sphere and can involve people deeply. Indeed, if an observer remains open to reception, as



he/she becomes engaged by the art, technical information could be communicated in a more effective way, and perhaps people would make choices more consistent with the underlying science. Therefore, we have combined three different forms of communication, science, music, and poetry, to convey the history and evolution of a landscape, aware that this experience has the power to amplify the beauty of a place and the desire to preserve it.

## 2 Objectives and methods

The purpose of this work is to facilitate a new perception of place, beginning with its beauty but ultimately arriving to a practical knowledge of its genesis, and of its problems and weaknesses which have influenced its cultural history.

A place must have the ability to attract for its beauty, nevertheless the understanding of the geological and geomorphological mechanisms that generated it allows people to fully appreciate the fragility of such beauty. Intimate knowledge might promote the desire of the public to protect a landscape, preserving it over time. Through experience comes an awareness, a connection, and even a love for the territory. Our goal is therefore to stimulate the interest of the public about the genetic aspects of the territory, since from these characteristics derive its beauty, its history, the culture and traditions that have developed around the place.

Our work proceeds through two different routes. The first step analyses the landscape from the scientific point of view trying to explain how it evolves and responds to changes in independent variables which are things such as climatic and tectonic conditions. This first step is accomplished using a simple scientific language, conceived for lovers of the territory of all ages and all cultural backgrounds. We convey the knowledge that the Earth has systems, sometimes in very delicate equilibrium that can respond in an unpredictable, complicated, and often disastrous way to the events or changes originating from geological processes such as climate changes, or human intervention. Providing the system context allows one to perceive the potential fragility in the environment.  The next step however is to establish a personal connection so that the public cares to further understand and appreciate the landscape.

This second step examines the landscape from a perspective more closely related to the visual and emotional impact that a place evokes: its history, its cultural significance, and perception of its fragility in a human context. The latter is perhaps a more abstract path, more intimate, which develops fully through the use of communication forms that exist to express human feelings, that is artistic language, such as music and poetry.

### 2.1 The working method

The sites were selected for their inherent aesthetic richness, but also for their different genesis and evolution, which result in very different visual impacts. A few years ago we created a team of five researchers-artists with different skills, called "TerreRare" (which means Rare Earth Elements but also refers to the rare lands of Italy), linked by a common interest for the region, whose mission is to promote a deeper public awareness of the landscape. To achieve this mission, we combined three types of "language": science, music, and poetry. We always begin by describing the geomorphological evolution of the place,



using a simple scientific language. Two geologists begin this story, analysing the processes and the "forces" that have created and modified the landscape over time. For each location, we indicate an itinerary with some stops, from which it is possible to enjoy particularly significant "glimpses" or panoramas, and from where we are able to narrate the peculiarities of the landscape. After some key words of the place have been identified, linked to its genesis, evolution, history or atmosphere that it evokes, we try to represent them through music and poetry. In other words, we try to translate the same information but following a

different, more direct emotional path. The musician, through the musical language, tries to reproduce, or harmonize with, the emotional impact of a site by searching for a piece of ancient music composed for harpsichord. The chosen pieces of music, as hereafter described, communicate aspects of the place through the elements that belong to the musical language. The choice of the musical instrument and the historical period is not accidental: the harpsichord has a punchy and gritty tone that clearly expresses the "strength" of the landscape. Early music, in addition, aptly suited to represent natural forms whose history began

millions of years ago; many late Renaissance or Baroque pieces have been composed to describe a specific situation, in fact, many pieces of music have a title. Other times the musical form (prelude, toccata, passacaille, rondeau, variation, canon, etc.) was critical in identifying the association with the place. In parallel, the poet expressly dedicates verses to these places, using powerful metaphors that become a cognitive tool linking nature and thought. Everything that in the realm of geological process might be shrouded in mystery is where the poet unlocks it, makes it available, lovable and palpable through a metric balance,

and studied cadences of heartbeats. Such analogies make you love a place, even parts unknown, not only for the great scientific interest, but also for the purely poetic fascination of all the things, which emerge from the deepness of time. Finally, for each site we suggest a section dedicated to one aspect of naturalistic, historical or cultural interest: a proposal to know "something more" around that place.

How are all these contents communicated to the public?

This project took place thanks to an important regional announcement dedicated to the development of the Marche Region. It started in February 2018 and we concluded in December 2019. The results of this work are offered to the public through different products: an important medium is a book just published (Nesci and Valentini, 2019), dedicated to 20 sites in the Marche Region (central Italy). The same contents of the book have been summarized and posted in a website (https://www.terreraremarche.it). The poems and the pieces of music, which you can enjoy individually for each place, are

also the soundtrack of videos that, by using beautiful images of these places, creatively interpret the science and nature through the art. These videos can be found in the website and in the DVD attached to the volume, where it is also contained the book in interactive form. However, above all, the contents of our project are offered to the public through live events, dedicated to the places.

**2.2 Description of the events**

The most effective and engaging communicative method of our work is through live events. Since the beginning of this project, we planned to address the public directly through shows of about one hour and half, that combine scientific communication (always by means of a simple and popular language) with the acting of poems and the performing of musical



pieces of ancient music. The project, as above described, include 20 sites from the region, but the shows were dedicated only to five or six of them, usually those closest to the place where the live event takes place. The events are structured in two parts:

the first one is carried out by means of verbal communication, with the employ of figures, sketch and schemes, trying to communicate the geological genesis and the geomorphological evolution of the places. It is not an academic lesson, rather the speaker uses a conversational language to involve the public, focusing on the most interesting aspect of the genesis, and investigating how the morphology of the landscape have influenced the history and culture of the place. The speaker guides the public and identifies some key words for the place. These key words will represent the link among the science, the music

and the poetry. The speaker, at the end of this part, encourages questions from the public. The second part of the events is a performance in *sensu strictu*: the actor and the musician are on a different plane with respect to the public and there is no interaction among the performers and the public, until the end of the performance. The lights are softened, and the shows develop. The performances are conducted in front of a large screen with projections of images and videos of the places: the result is a total emotional immersion of the public in the place. Up to now, we have organized about ten live events, all in

central Italy, some of them in the proximity to one of the sites. The performances were carried out in different locations, but always searching intimate and suggestive places with acoustic features suitable to the kind of the music (ancient, by using the harpsichord). Among these locations, the most suggestive were probably the church of S.Maria in the charming bay of Portonovo (Ancona), one of the most spectacular examples of Romanic architecture of central Italy; the theatre of the prominent Renaissance Fortess of Sassocorvaro (PU); the "Sala del Maniscalco" part of the magnificent Ducal Palace of

Urbino (PU). We have in program also outdoors events, but they have not yet been realized. Moreover, we hope to organize more events outside of the Marche Region and also from Italy, with the aim to attract people in a territory that still has a great development potential, still uncontaminated and rich in nature, culture and traditions.

### 2.3 Experience with the public

In the events, we always distribute a program of the show, indicating the localities that we are going to illustrate, the text of

the poems and a track about the pieces of music chosen for each site; also, we add our contacts in the program. Until now, we have never asked for a written feedback from the public. Our experience, therefore, is based on the direct feedback from the public: the evidence of interest shown, the questions we are asked there, the requests for further information we receive by email, the interest shown in the YouTube channel and in social media. The involvement of the public during the shows was always very high, and the number of participants was also satisfactory, even if it's difficult to quantify these data, because this

value depends strongly from several variables. For instance, it depends from the advertising made before the event but also from the season and the weather, the beauty of the place and how difficult is to reach it. Is also important the receptive capacity of the performance hall and the context of the event (if it is, or not, linked to a group of events organized that day in the place). We underline that the project is now at a crucial step: we have just completed the book and we are improving the web site. This material represents a fundamental step in promulgation and dissemination of the project. At this point, we have the

concrete material to propose the events to the public and collect the response of the participants.





We emphasize that this work can be proposed to people of different ages and cultures. We have not done performances in didactic contexts (school, museum), but the presence of students of different ages during the shows encourages us to move in in this direction also. Our events are addressed to a general public of all ages, an assorted and wide audience, people interested in the territory, or in the poetry or in the ancient music. The show is not conceived for scientists even if, at a geomorphological

meeting in Urbino (PU, Marche Region) it was much appreciated by the participants as a cultural moment. However, the most satisfying result is related to people who show interest in those aspects of landscape evolution that affected the morphology of the place and consequently, the history of human settlements and culture.

## 3 The three case studies

As a description of case studies, among the many amazing landscapes of Italy we focus on three charming sites from the

Marche Region (Fig. 1): "The sea-cliff of San Bartolo", "The flatiron of Mount Petrano" and "The fault of Mount Vettore".

### 3.1 The sea-cliff of San Bartolo

### 3.1.1 The geology

The northernmost sector of the Marche Region is characterized by a high and rocky coast that interrupts the continuity of beaches that fringe the Adriatic Sea south of Trieste. The sea-cliff of Mount San Bartolo (Fig. 2) is more than 200 meters high

and represents the outermost ridge of the Apennine chain, which extends to the Adriatic Sea. The natural beauty of Mount San Bartolo, makes it home to a natural park very popular with tourists during the summer season (Savelli et al., 2017). The outcropping late Miocene rock formations are represented by marls, marly limestones, dark  mudstones and bedded sandstones and marls. The small and sporadic pebbly beaches that protect the base of the relief are eroded by sea waves during the strongest storms. Instead, the less protected rocky ridges are directly attacked by the waves (Fig. 3). For this reason, but also because of

the easily erodible lithology and the fracturing of the rocks, the slope facing the sea is affected by extensive landslides that endanger the overlying villages of Gabicce Monte, Casteldimezzo, Fiorenzuola di Focara, Santa Marina, and all of the panoramic road (Fig. 4). The natural causes of the instability of the mountain are superimposed on the anthropic ones: in fact, man has intervened heavily and disturbed the precarious balance, accelerating erosional processes. One of the most important causes of coastal erosion is the decrease in the sedimentary contribution by the rivers, which flow southeast of the relief. The

solid load of these rivers, in fact, contributes to feeding the Pesaro beaches as the coastal currents transport the sediment to the north. The solid flow of rivers has significantly decreased due to the construction of dams on rivers and due to the extraction of aggregates from the riverbeds (Colantoni et al., 2004). To stop the advance of the sea and reduce the risk of landslides, several barriers have been built which, while slowing down the process, have not solved the problem of the erosion of the relief and corrupt its wild beauty. The natural response of the coastal system has been the unnatural formation of sandy beaches

between the cliffs and the shoreline, other than the triggering of complex refraction and diffraction processes of the wave





motion on the breakwater structures, which determines dangerous coastal currents. The retreat of this stretch of coast has been fast and continuous since the Holocene (11,000 years ago) and the shoreline has changed continuously over time. The paleo-coasts are now erased even though there is unmistakable geomorphological evidence of their existence in seaward locations of the submerged beach. Geomorphological data testify that about 6000 years ago, the paleo-coast compared to the current one

was advanced of about two kilometres and the relief of San Bartolo was much more extensive and stretched out towards the sea (Nesci, 2003).

Here we propose a route that leads from Fiorenzuola di Focara, an ancient village balanced on the cliff (Fig. 5, stop 1), to the beach below. Along the path, the thick and dense sequence of marly and clayey layers of Mount San Bartolo can be observed and the precariousness of the rock-wall on which the village rests is perceived (Fig. 5, stop 2).

How to communicate the peculiarities of this landscape through poetry and music? Our main objective was to identify a key to interpretation, some key words, which synthetize the main genetic as well the process mechanisms, other than the critical issues: stratification, balance, fragility.

### 3.1.2 The Poem

The title "Up / Down / Fragile" recalls the "handle with care" written on the packaging of fragile objects. In a sense, Mount

San Bartolo should not exist, so much is the stress placed on it as it undergoes a continuous consumption by guzzling by the jaws of erosion, of human development, and of economic profit. Yet in the moment, it remains in perfect balance just by virtue of its beauty. The visual balance (color), olfactory (scent) and physical (sky and sea) is represented by the warmer and motionless hour of the summer season ("hour without a shadow"). To balance the Mount is the set of ecstatic looks and love that history, despite everything, offers. Appreciate this beauty says the poet, for it is fragile. Below is the poetry in the original

language and its translation.

**Alto/basso/fragile**

Oblio di un'ora senz'ombra
   Occhio distratto alla Storia
     Dove per eritemi ed ustioni
215       Per strappi graffi ulcerazioni
        Oltre /memorie/ di imboscamento
        cede finalmente la mano
     (al tuo) essere nonostante
     (al tuo) essere sacro Monte
inspiegabilmente te-stesso testimone
del tuo esistere per sottrazione



In questo tuo

ALTO                              BASSO

225                     FRAGILE

Dove ogni strategia umana

        Mai è apparsa così vana

            È vanità di utilissimo umano andare

230                 Di avido & arido e vacanze week end e utilitarie

                    È il brusìo giallo delle tue ginestre

                        Impossibili a categorizzare

                        E poi tu

                    dispendioso

235                 ancora tu o Monte

            orsù (scendi) è ora di andare

        sù sù Monte è la tua ora

è una questione tra te e il mare.

**Up/down/fragile**

Now is the time:

        forget your shadow

            steal your eye away from History

                sneak through scalds and burns

                    snatches and drills then overturn

245                     where are your /memories/ ambushing

                    surrender at last

                (to you) being despite

                (to you) Holy Mountain

            unexplainably yourself

witness of your existence by abstraction

            This is you:

UP                              DOWN

            FRAGILE




just where each human strategy game

    never has been so vain

        vain is the valuable man

           vain is the eager vain is the barren

260              vain's out for the weekend on a cheap city car

             But there: the never classified

        whispering of your brooms

      And then you

    lavish

you again Mountain

  come on (come down) we must try to go right now

 hurry up, Mountain, it's time

the matter's between you and the sea somehow

### 3.1.3 The Music

The piece of music selected to represent this site is the Passacaille from Suite VII in G minor HWV 432, by George Friedrich
Händel (1685-1759). The Passacaille is a folk dance of Spanish origin, and summarizes in its main structure the morphological
appearance of the San Bartolo cliff and, besides, the poetic interpretation by the writer of the poem. The Passacaille, indeed,
is made up of variations on a ground bass; the same sequence proposed in a varied repetition, following accurate rules of
composition. That is exactly like the repetition of the layers on the cliff: layers that are in balance, although at times surprisingly
unstable, on the previous. This piece of music is bright and overwhelming; as the San Bartolo cliffs are an explosion of colours
and vitality. In the last variations, several repeated arpeggios, like a series of sea waves on the cliff (https://youtu.be/-
2a4i6iOGE0).

## 3.2 The flatiron of Mount Petrano

### 3.2.1 The geology

Mount Petrano has a relief of 1162 m, which stands out in the landscape for its characteristic flattened top visible even from
long distances. The gorges of the Burano and Bosso streams, which cross the mountain ridge transversely (Fig. 6), separate
the Mount Petrano from Mount Catria to the south and Mount Nerone to the north. The panorama that can be enjoyed from
Mount Petrano embraces a large territory offering spectacular views of the major peaks around it, and on the whole Province
of Pesaro and Urbino, up to the Republic of San Marino and the Adriatic Sea. The relief constitutes a beautiful example of
anticlinal ridge (Nesci et al., 2005). Anticlines are one type folded of rock layers, formed by compressive forces that rise slowly
over a very long period. But folded rocks are intriguing, hard to imagine and yet there they are, in plain sight. The morphology





of relief perfectly follows the wide anticline and no tectonic dislocation seems to disturb the simple regularity of the fold (Fig. 7) exposing the anticline in its full and awesome natural wonder. Then too, the sedimentary rocks that outcrop in the ridge are alternately more or less erodible. The large smooth and almost flat upper surface consists of carbonate rocks of Maiolica
Formation, highly resistant to degradation. The formation of Marne a Fucoidi, which rests on Maiolica, are more degradable, being formed by marl and clays. Still above, the rocks return to being more resistant being formed by Scaglia Bianca and Rossa hard limestones (Alvarez, 2019). The selective erosion caused by run-off waters promotes the formation of the so-called "flatirons" which are subtriangular prismatic forms that recall the tip of an iron, from which their name derives. The observation of these spectacular forms is almost like being inside a natural laboratory, where the public gets to observe folded rock formed
underground, somehow raised to the surface, and then sculpted by streams on the surface.  The scale is overwhelming, the forces involved immense, and the resulting landscape is a work of art. The streams running down the mountain from the flat top of Mount Petrano, follow paths related by the degree of erodibility and fracturing of the rocks, finally producing these characteristic morphologies. The flatirons are very well exposed on the sides of Mount Petrano, even if they represent forms that are quite common to other anticline ridges of our Apennines; the most significant in this area is the one called "La
Roccaccia" (Fig.8). At the latter is genetically linked the small relief of "La Rocchetta" (1163 m), which rises above the structural surface of Mount Petrano (Fig.7). This little hill represents a remnant of the ancient structure removed by the erosion. For this site, we propose a route at the top of the Mount Petrano (Fig. 9). The path surrounds the hill of "La Rocchetta" and allows you to climb to the top of this hill.  From there, you can benefit from an impressive 360-degree panoramic view over much of the northern Marche Region and part of the Umbria Region (Stop 1). Finally, reaching the stop 2 you can observe, in
all its magnificence, "La Roccaccia" flatiron.

Again, the same question. How to communicate the peculiarities of this place through art? Here, the interpretation keys chosen to communicate the place through poetry and music were selective erosion, geometric shapes, childhood games.

### 3.2.2 The poem

The vision of pointed shapes on a plain dotted with lawns and flowers reminiscent one day holiday, cries of children playing
and drawing. On joyful suspension of play, the reality, far away and top view, burns and disappears in the flames.

**Picnic**

Felice di accettare l'invito
                a nozze
                    a tempo
315                     a luogo
per la casetta di fata di fiaba ^
felice convito sul tetto ^
            partita





pic-nic

ovetto sgusciato di nuovo

*«Mentre dall'alto possiamo ammirare*

*il moderno complesso andare in fumo»*

_ _ _ _ _ _ _ _ _ _ _ _

**Picnic**

I revel in coming here

on time

on place

a fairy tale hut ^

the merriest banquet  up on the roof ^

scrimmage

picnic

wriggling by this little lovely egg

*"While down there we can admire*

*the modern complex caught by fire"*

_ _ _ _ _ _ _ _ _ _ _ _

### 3.2.3 The music

Twelve Variations in C major on the theme "Ah, vous dirai-je, Maman" KV 265, is a keyboard composition by Wolfgang
Amadeus Mozart (1756-1791), published in Vienna in 1785 and probably composed when he was around 25 years old (1781 or 1782). This piece consists of 13 sections: the first one is the theme, the French folk song "Ah! vous dirai-je, maman". This French melody first appeared in 1761, and has been used for many children's songs, such as "Twinkle, Twinkle, Little Star", "Baa, Baa, Black Sheep" and the "Alphabet Song". This traditional piece deeply fascinated Mozart, who took up the theme in a playful composition. The following twelve variations (in rhythm, harmony, and texture) are developed in a very simple way,
to produce funny transformations that gradually articulate the starting melody, like several pieces of a children's game. The simplicity of this paradigmatic example of musical variation reflects effectively the idea of geometric shapes in childish drawings, or a toy of building blocks (https://youtu.be/RYS5-25rulk).



### 3.3 The fault system of Mount Vettore

### 3.3.1 The geology

Central Italy was struck by strong earthquakes in 2016 (Aringoli et al., 2016). On August 24, a magnitude 6.1 event shook the area between Marche, Lazio, and Umbria, devastating the villages of Accumoli, Amatrice (Rieti), Arquata del Tronto and Pescara del Tronto (Ascoli Piceno, Fig.10). A second quake occurred on October 26 with magnitude 5.9 and epicentre between Mount Cardosa, Castel-santangelo sul Nera, Ussita and Visso (Macerata). Four days later, on October 30, an even larger event (magnitude 6.5) destroyed the town of Norcia (Perugia). With the choice of this site, we wanted to instil a "beauty shock" to

infuse positive energy, break the darkness of destruction, and rekindle the life and creativity in a land rich in geological heritage and artistic masterpieces.

The Castelluccio di Norcia basin is a closed depression of over 12 km in length and 8 km in width, located in the southern part of the Umbria-Marche Apennines and surrounded by an uninterrupted mountain ring. It consists of three closed plains: Pian Grande, Pian Piccolo and Pian Perduto. The Pian Grande (Fig. 11), five km long in the NW-SE direction and two km wide, is

the largest and most spectacular one, and it is located to the west of the alignment of Mount Vettore – Mount Priora.

To comprehend the genesis of this area, it is necessary to understand the formation of the Apennine chain that consists of a compressive thrust system towards the Adriatic Sea. Toward the Tyrrhenian side, in the innermost part of the Apennine compressional system, an extensional zone occurs. Just in this sector, large and elongated intermontane plains have formed, which are bordered by complex systems of direct faults and fractures. Imagine a bulldozer that shoved layers of the upper

crust and then when it stopped pushing, some of those layers started to slide backwards. But there was no bulldozer, just an Earth system, evolving, whether you are watching it or not. And yet you can't avoid "hearing" this evolving process, because you can still feel the rumble of this process which we call earthquakes. Our site captures a moment in this process.

The Castelluccio plain was not always a closed depression, in fact during the Middle Pliocene - Lower Pleistocene (about 3.5 Ma) the subtropical humid conditions created a landscape with low energy relief, a large paleo surface. The subsequent

distensional tectonic phase characterized by the reactivation of direct fault systems with NNW-SSE trend and by an intense uplifting of the area, has strongly dismantled the paleosurface preserving it only in reduced fragments in the ridges. These processes interrupted and disarticulated the previous landscape, forming a series of tectonic depressions that characterize the whole Apennine area. A bundle of Quaternary faults of Mount Vettore - Mount Bove to the east and of Mount Castello - Mount Cardoca to the west surrounds the depression of Castelluccio. This depression is related to an extension direction, oriented

around NS, which led to an oblique movement along the main border faults, and to the development of the tectonic depression of Pian Grande (Passeri, 1994).

Starting from the end of the middle Pleistocene, the depressions were filled with debris deposits coming from the slopes through the short streams. Their placement is due to the glacial and periglacial morphogenesis as evidenced by the presence of large cirques and glacio-nival niches in the surrounding areas. The flat surface of the Pian Grande is affected by sinkholes

in which the surface waters are dispersed in depth (Aringoli et al., 2018).



To the east of the plain stands Mount Vettore that, with its 2476 m height, is the major topographic relief in the whole Marche Region. Even the most inexperienced eyes cannot miss the large scar that cuts the mountain (Fig. 12), the direct fault that displaces the Sibillini's thrust plane. This fault appears in all its majesty at the base of the Aquila rock, consisting of massive limestone. Other faults are less visible, for example, the huge fault at the base of the mountain associated with the formation
of the plain and many fractures scattered on the side. The whole slope is strongly deformed by landslides, both deep and superficial, and by some fractures reactivated by the earthquake.

Mount Vettore represents one of the most popular destinations for excursionists in the Marche Region. The proposed route, (Fig. 13), starting from "Forca di Presta" (1550 m) until crossing the fault of Mount Vettore, can be, for trained and experienced walkers, only the first part of the longest and most difficult itinerary that leads to the top of the Mount Vettore (2476 m) or to
the beautiful Lake of Pilato (1941 m). The route is always well signposted; the difficulty is medium, due to the steepness of the slope and the coarse gravel on which it is not easy to walk. Enjoy the magnificent view of the Castelluccio Plain at Stop 1, and then continue towards the top of the Mount Vettoretto (2032 m). After about an hour of ascent, the fault of the Mount Vettore appears impressive, on your left. The fault plane is clearly visible, exposed for about 2.5 meters (Fig. 14). A little higher up, the route crosses a series of fractures resulting from the 2016 earthquake, some of which form open the ground
fractures for a width of 30-40 centimeters (Fig. 15).

Here, the interpretation keys chosen to communicate the main peculiarities of this place through poetry and music are; fault system, readjustment, balance.

### 3.3.2 The poem

The spectacle of the suspended mountain, in precarious balance, captures the attention and forces you to breathe with your
gaze turned upwards, with an apparently unnatural rhythm. The natural cycle of nature tends to plan, to the "horizontal world", but some events are opposed, and this creates tension, even emotional. Reading backwards from the last verse to the title of the poem emphasizes this "return and stay" and accompanies the gaze and the contemplation.

**Funambolo**

> ansioso
> all'altrui sguardo
> appeso unicamente
> torni e resti,
> in perenne squilibrio
> tu come me
> monte funambolo,
> vorrei raggiungerti
> Nel tuo regno orizzontale





**Tightrope walker**

anxious gaze

to another Man's

hung

you go back and stay,

in everlasting imbalance

you, just like me

tightrope walker Mount,

I would like to join you

In your horizontal kingdom

### 3.3.3 The music

Johann Sebastian Bach (1685-1750) - *Canone per Augmentationem in Contrario Motu* (from *L'Arte della Fuga* BWV 1080).
The basic principle of the canon is the imitation. In the canon, a melody is faithfully repeated by the various voices in regular succession. An elementary example, well known to all, is Fra' Martino campanaro (Brother Martin), whose melody is always the same, repeated in the other voices in subsequent times. That is the case of a canon in unison, while more interesting is when the main motif is repeated at different heights from the initial one. The movement of the main voice is "the law", and is repeated in the other voices, at different heights, or reversed (retrograde canon), at the mirror (inverse canon) or with different
rhythmic measures (mensuration canon). The absolute summit in the construction of the most complicated canon compositions belongs to Johann Sebastian Bach. In his masterpieces *L'Offerta musicale* and *L'Arte della fuga*, the most interesting examples appear: infinite, retrograde, inverse, retrograde-inverse, mensuration and enigmatic canons. The *Canone per Augmentationem in Contrario Motu* here proposed, is a perpetual canon with two voices where the second voice repeats (at the lower fourth) the same melody but exposed for opposite motion (specular) and with notes of doubled value.
As in the canon the melody is differently repeated in the various voices, in a fault system the movement along the main fault is followed by readjustment along the other faults in the surrounding area. The analogy among the mechanisms of the canon and a fault system is very strong. It is a wide, intermountain basin, delimited by several faults, mostly distensional, with different orientations but strongly linked together. The fault of Mount Vettore is there, concrete and clearly visible. To its movement the whole system reacts accordingly, until reaching new equilibrium (https://youtu.be/nXZGPwJDdAc).

### 4 Discussion and Conclusions

This work presents a new multidisciplinary and interactive method, which seeks promote the communication of rigorous and complex scientific content, related to the geological genesis and the evolution of a landscape, through art forms such as poetry and music. Art has a great power in-and-of-itself and in the communication of specific subjects, an opportunity not to be





overlooked. By addressing the emotional sphere, art manages to engage the observer in a profound and passionate way. The communication of information of any nature through the emotional sphere is recognized to be much more effective than traditional communication methods. Our experience can confirm this effectiveness: attendance by a very large and varied audience, mostly without a scientific background at our live shows uphold a great interest in the problems proposed. The published volume, the DVD with the same contents of the book in interactive form and the material posted on the web site, which were finalized at the end of the 2019, represent an important step of the project. The next one, probably within one year, will be able to consider the response of the public relatively to these different proposing ways.

This work proposes small real or virtual trips to the Marche region: it is possible to go to places, follow paths and stop there to listen to the piece of music, read and listen to poetry. Alternatively, you can follow the route from home, using the virtual mode, listening to music and poetry while watching the videos of the places. It is possible to visit the website, where the contents of the book are summarized; one can participate in live shows, periodically organized in various locations, within and outside the study region (the site contains information on the dates and places of the shows). The performances are organized following the multidisciplinary communication method above described, combining science, projections of images and videos, recitation of poems and live musical performances, with the aim of generating an emotional fusion that introduces the public in the strongest and most engaging way inside the wonders that our land offers. The aim of this project is not to provide yet another guide to visiting a beautiful region. The project is in fact much more ambitious, since it wishes to stimulate love for the territory through knowing it, in its formative and evolutionary mechanisms. From this knowledge, indeed, it is possible to understand how nature, human settlements, history, culture and the traditions of a place have developed. This method addresses the curious, the lovers of the territory and art, to those who want to get to know "the place" in all its aspects. People of all ages and backgrounds can participate in this experience, to stimulate in them an awareness of which cultural heritage the landscape represents, with the aim of increasing their desire to understand the fragility of the territory to finally stimulate the protection and conservation of this heritage. The landscape must be understood to be loved and protected. We have the scientific responsibility for the conservation of the priceless landscape heritage of our Earth for future generations.

**Author contribution**

The authors contributed equally to the research and writing of the manuscript.

**Acknowledgements**

Thanks to Prof. Larry Mayer for the English language revisions and suggestions. Thanks to the Marche Region for supporting the project from which this work originates.



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






**Figure 1: Digital elevation model of the Marche Region with the sites involved in the Project. In red the three case studies analysed in this paper. ©2020 Marche Region.**





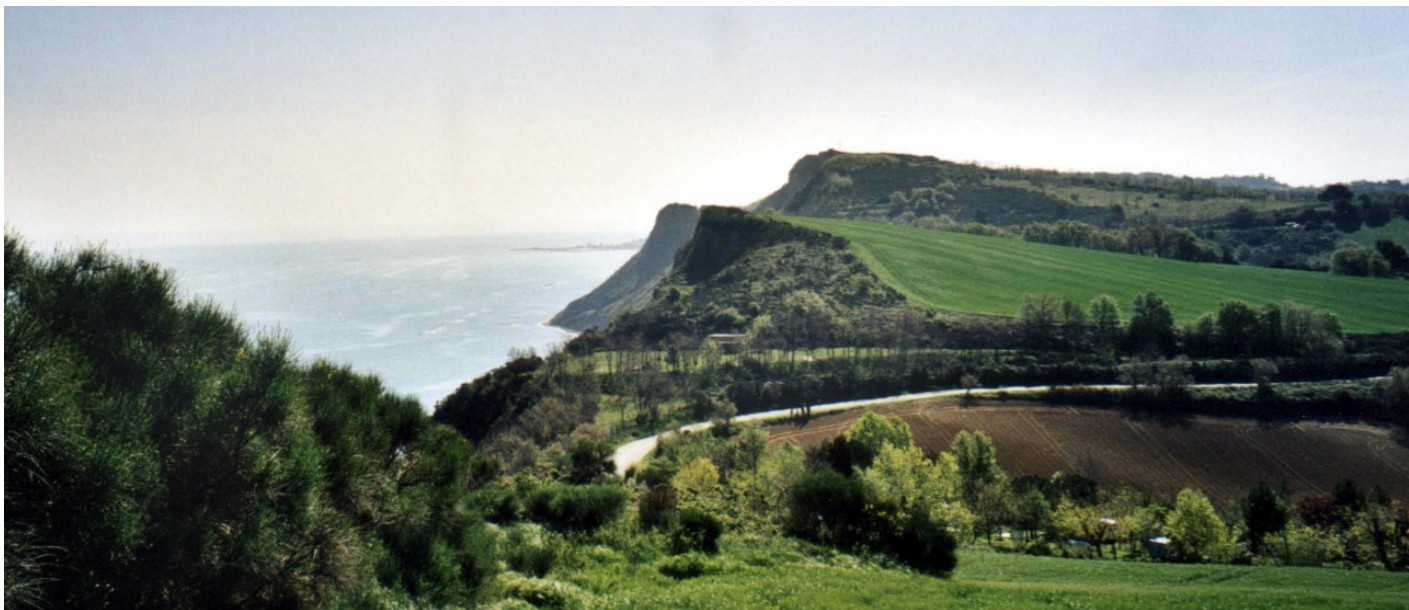

Figure 2: Panoramic view on Mount San Bartolo.





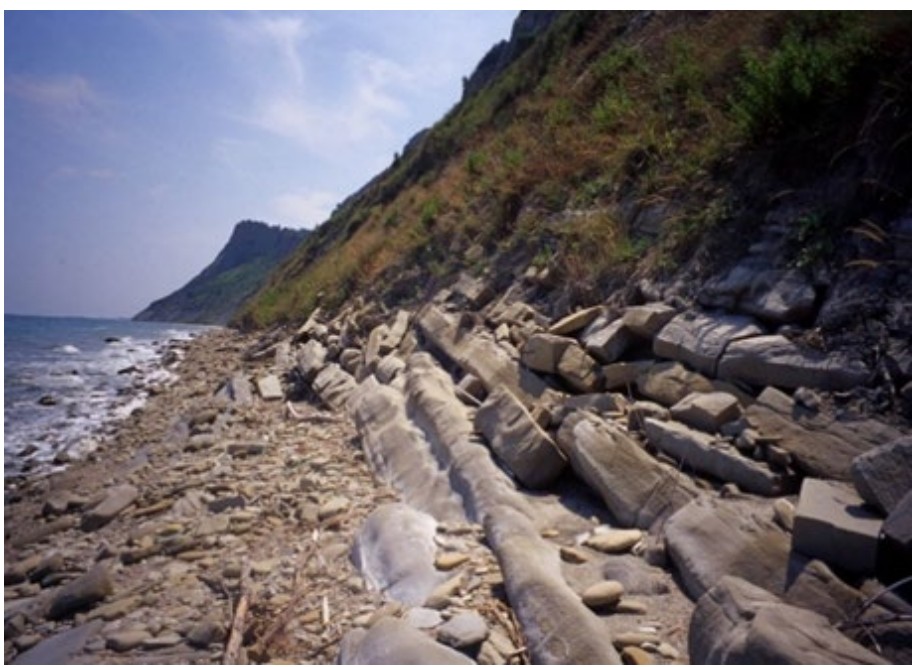

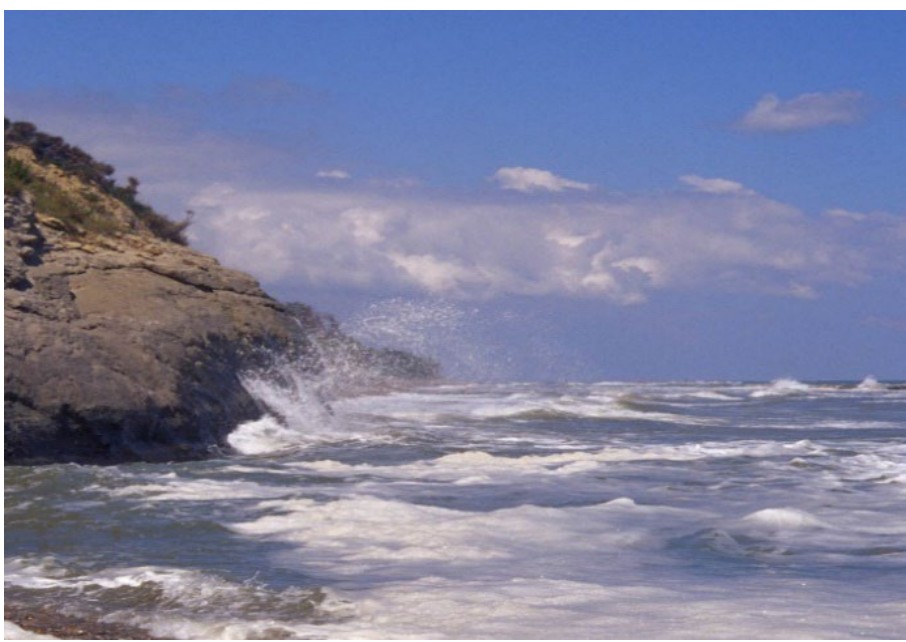


**Figure 3: Pebbly beach (up) and substrate eroded by waves (down).**





**Figure 4: Panoramic of the active cliffs of Mount San Bartolo.**



**Figure 5: The proposed itinerary. P = parking area; 1 and 2 = stop. ©2020 Marche Region.**





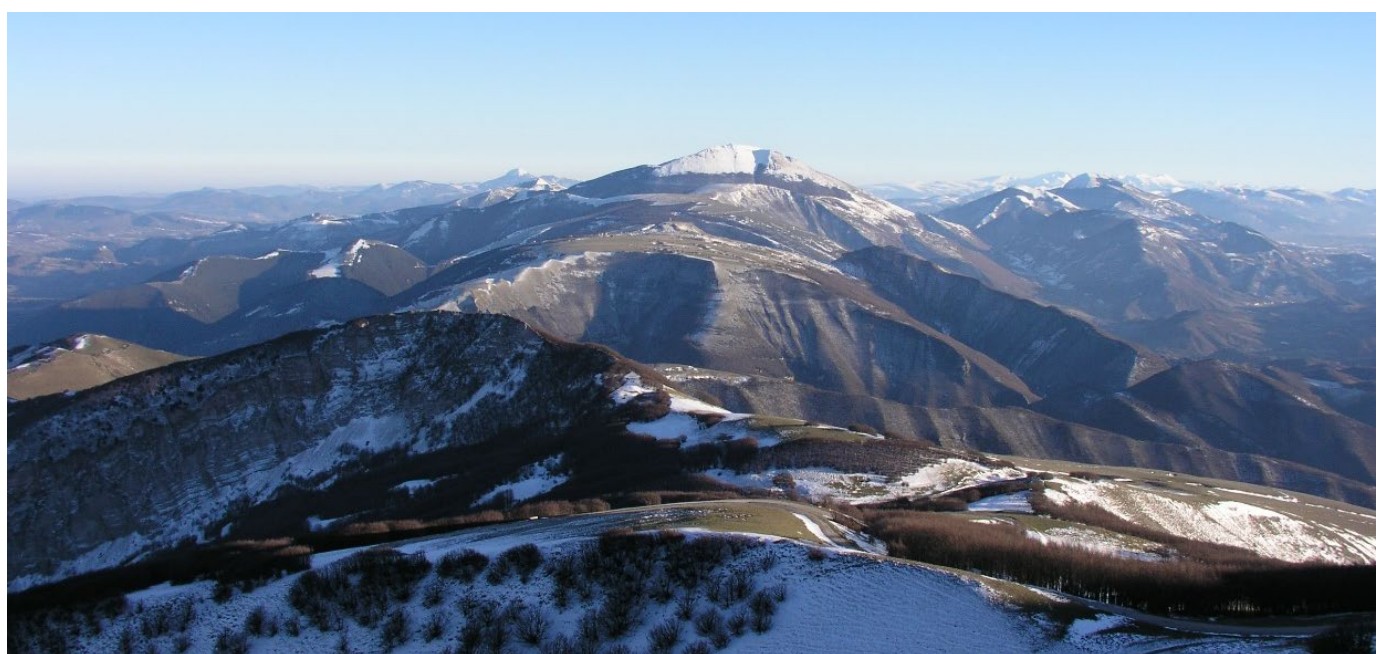

**Figure 6: The Apennine ridge seen from Mount Nerone. Mount Petrano is located between the incision of the Bosso and Burano streams. In the background you can see Mount Catria.**



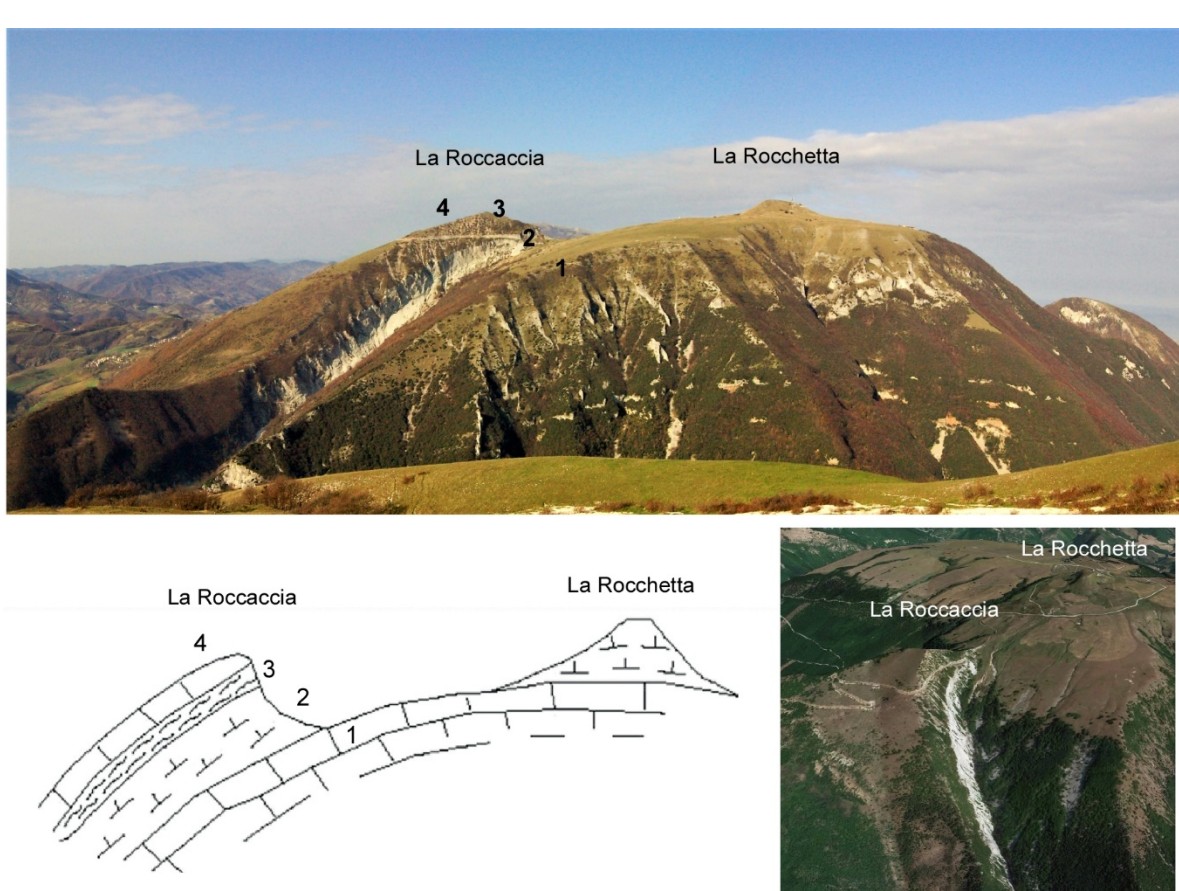

**Figure 7: Panoramic view of the Mount Petrano anticline and illustrative sketch of the relief morphology that was formed by selective erosion. 1. Maiolica Fm, 2. Marne a Fucoidi Fm, 3. Scaglia Bianca Fm. 4. Scaglia Rossa Fm). Bottom right: aerial view of "La Roccaccia" Flatiron (Map data ©2018 Google).**



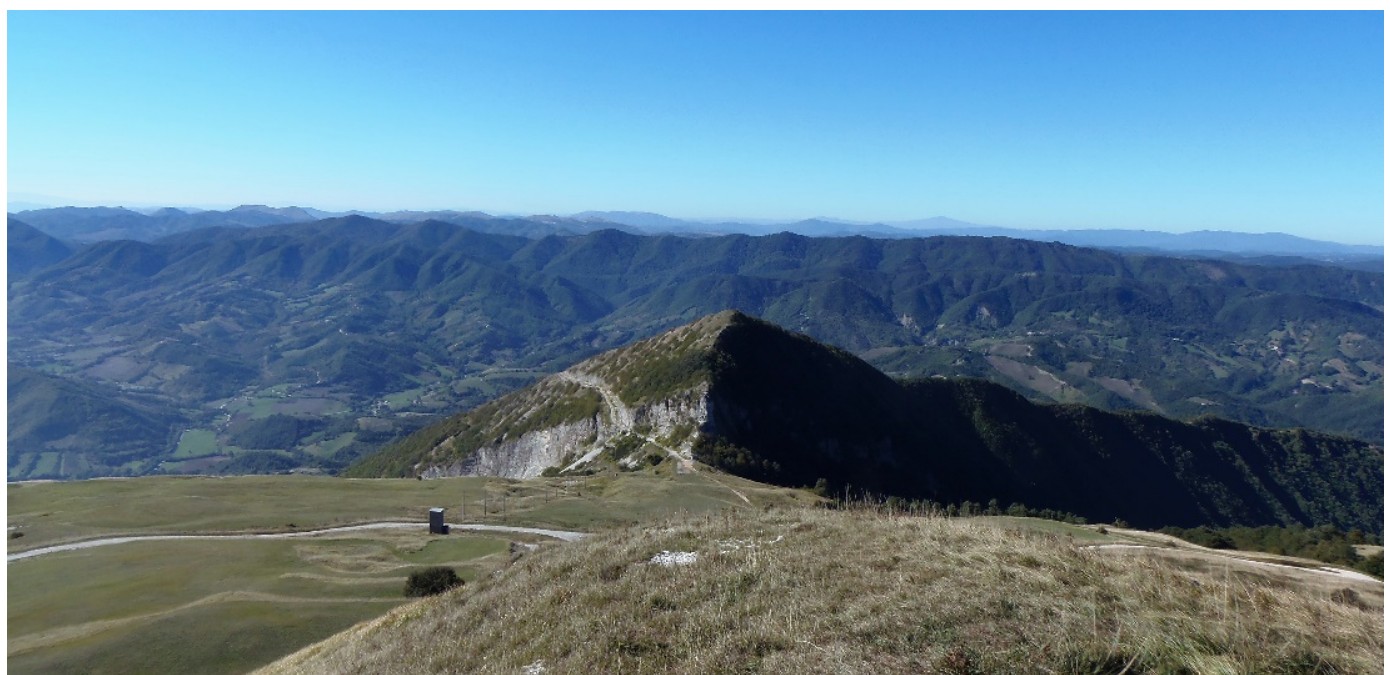

**Figure 8: "La Roccaccia" flatiron.**







**Figure 9: The proposed itinerary at the top of the Mount Petrano. P = parking area; 1 and 2 = stop. ©2020 Marche Region.**



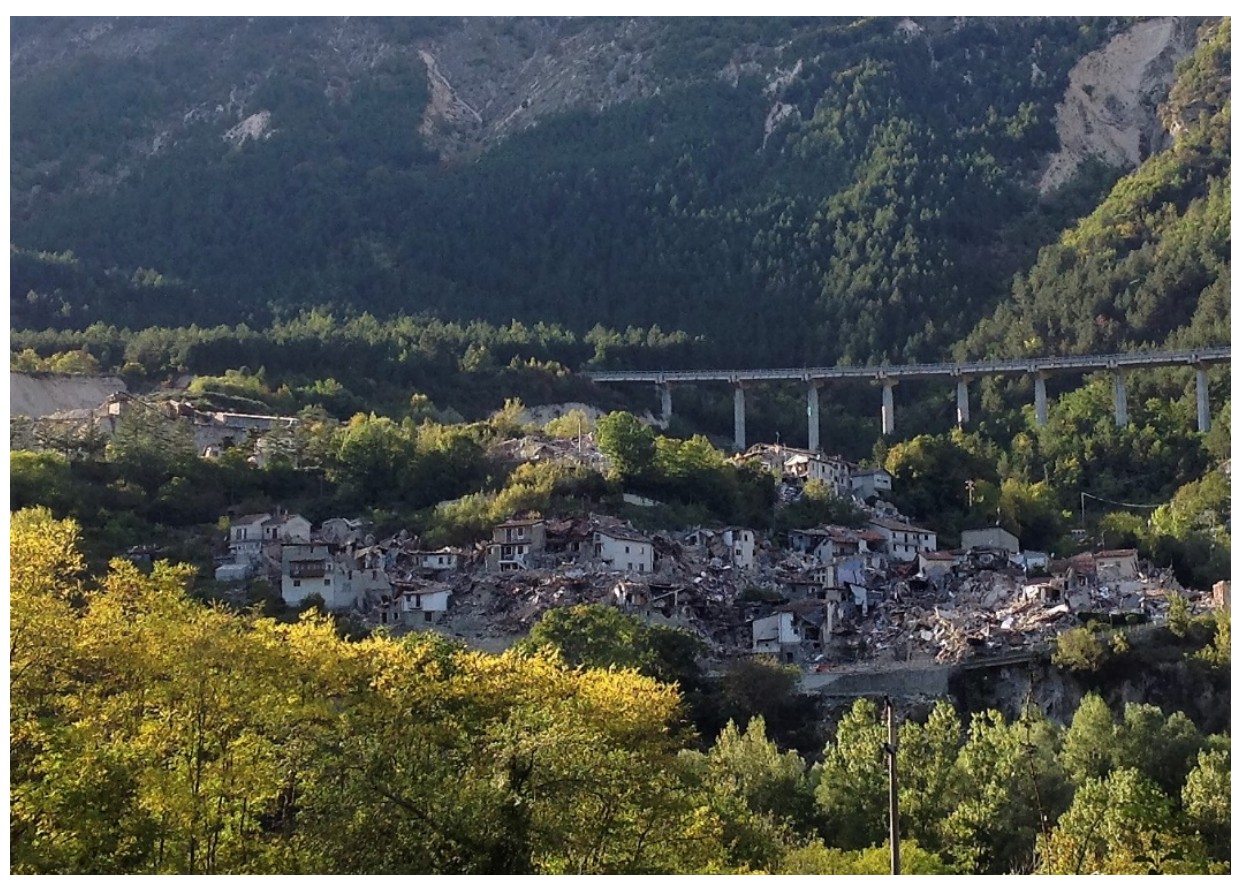

**Figure 10: The town of Pescara del Tronto razed by the earthquake.**

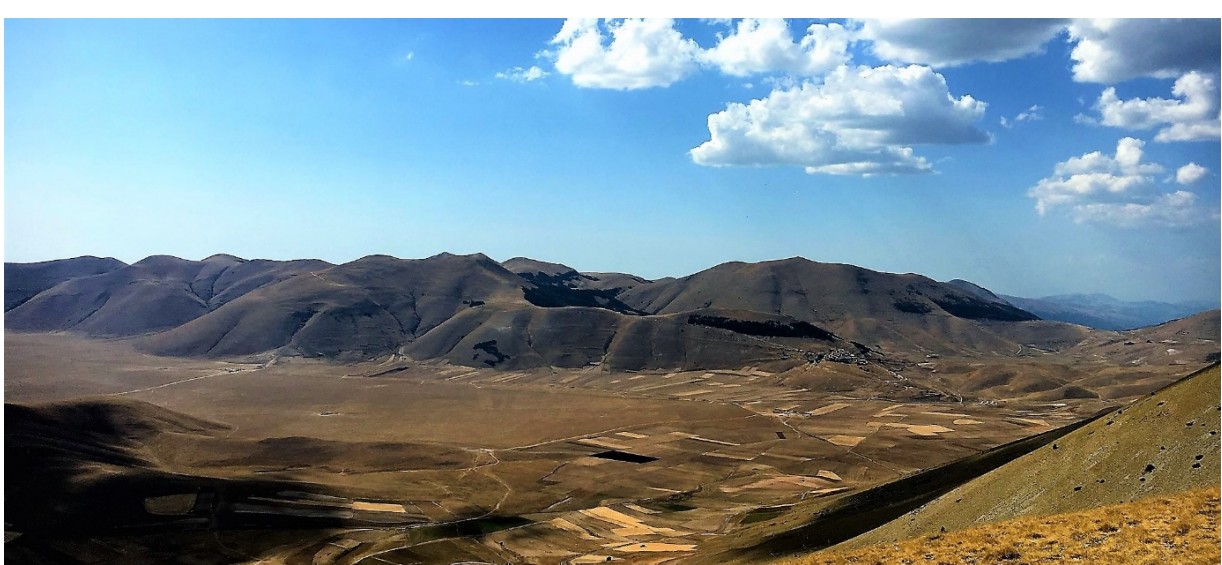

**Figure 11: View of the Pian Grande.**




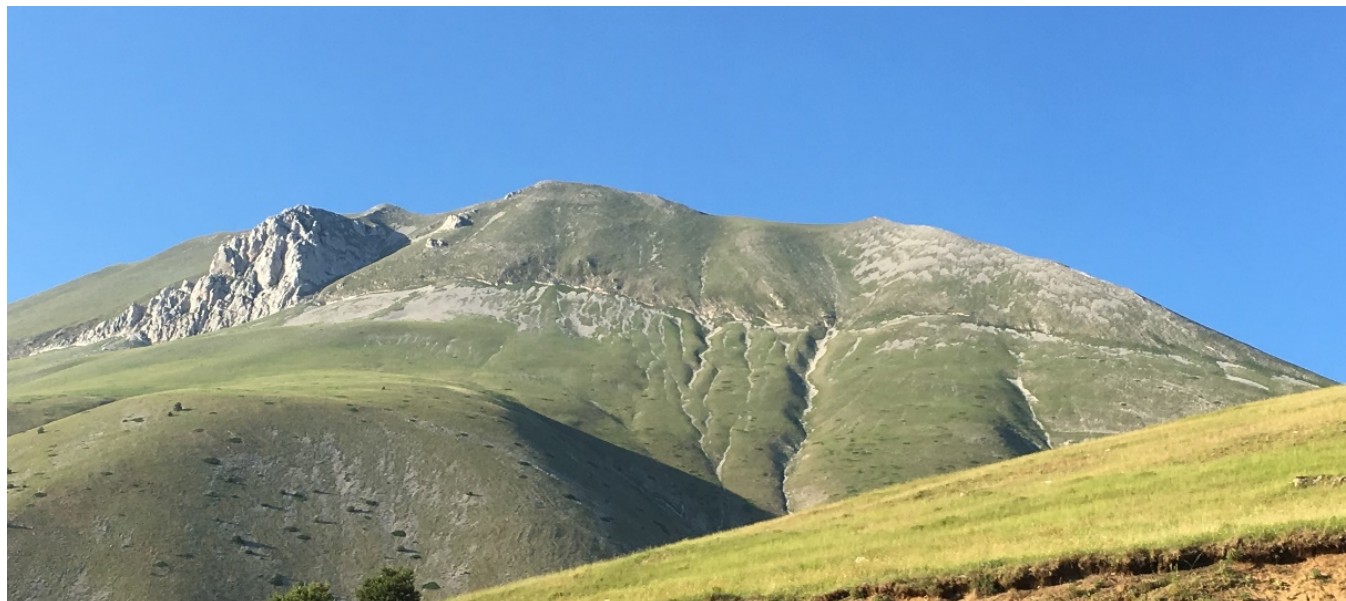

**Figure 12: The fault at the base of the L'Aquila rock.**





**Figure 13: The proposed itinerary at the top of the Mount Vettore. P = parking area; 1 and 2 = stop. Blu dashed line = the fault.**
**©2020 Marche Region.**



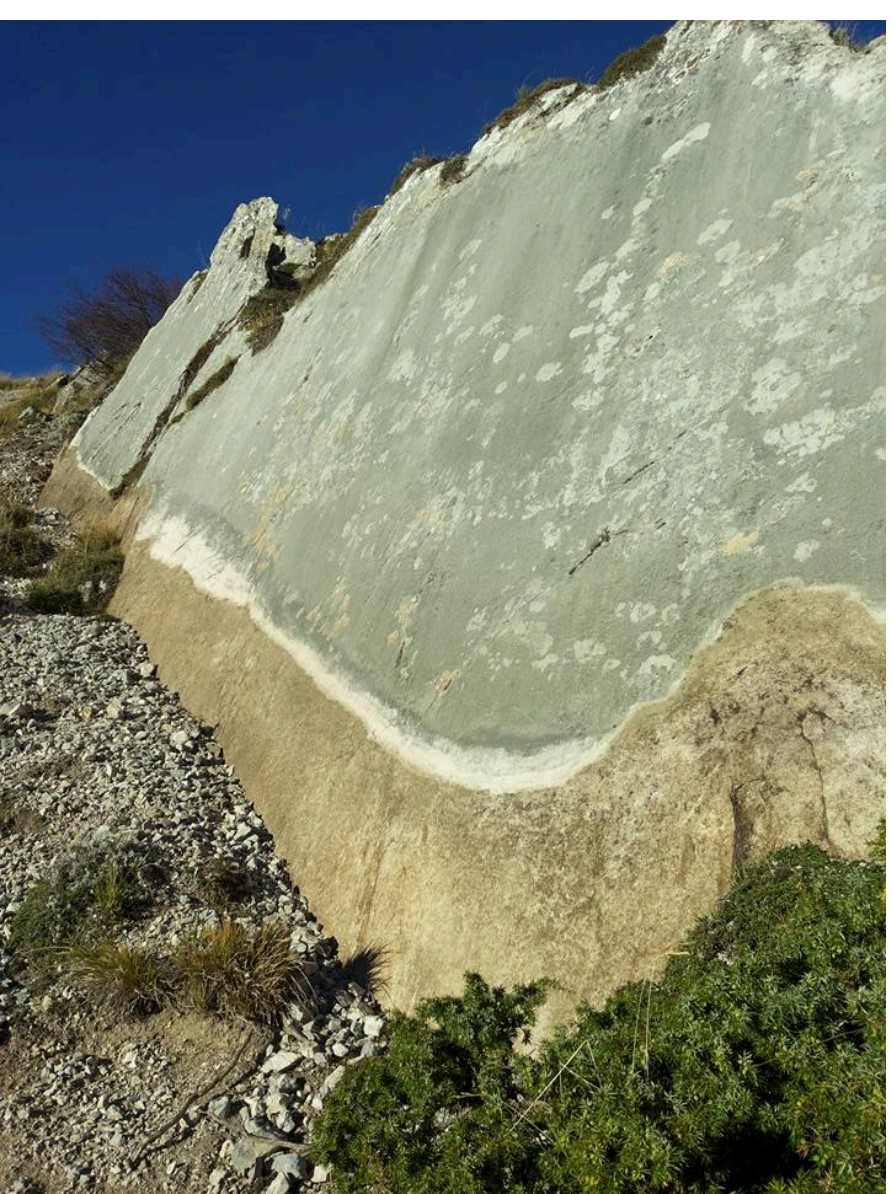

**Figure 14: The fault plane.**





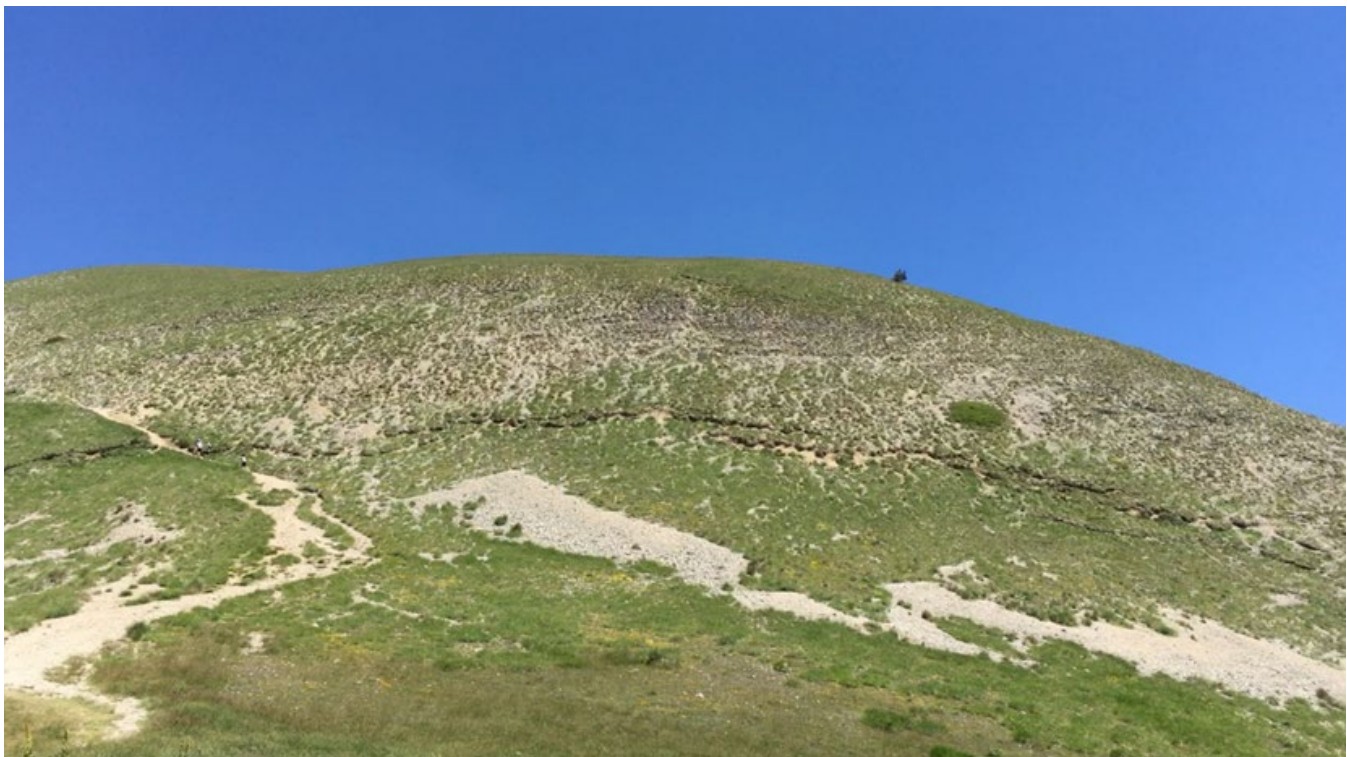

**Figure 15: Some fractures related to the 2016 earthquake.**