# Peer review of "Science, Poetry, and Music for Landscapes of the Marche Region, Italy: Communicating the Conservation of Natural Heritage"

_Geoscience Communication, 2020_

## Referee Comment (RC1) · Mauro Soldati (Referee) · 5 May 2020

General comments

The paper aims at describing how geological and geomorphological landscapes has been bridged with Art disciplines in the Marche Region (Italy) through an innovative project and stimulating initiatives addressed also to non-specialists. The topic is certainly suited for publication and discussion in Geoscience Communication. The structure and content of the paper are of good quality. The manuscript would require just minor revisions which are suggested below.

[Figure]

Specific comments

The caption of the figures require an effort to make them self-explanatory. In their present form, most of them appear too brief to fully explain the content of the images. Enriching the captions would make the whole paper more informative and attractive, especially for readers who are not familiar with the investigated areas. Specific suggestions about single captions are provided below, together with some punctual comments and suggestions regarding the whole text.

Line 142: 'S.Maria' should read 'S. Maria' (with added space).

Line 145: 'We have in program also' should better read 'We are also planning' or 'We have also planned'.

Line 165: The acronym PU may not be understandable for non-Italian readers.

Line 165: A comma would be needed after the parenthesis.

Lines 166-167: The expression 'aspects of landscape evolution that affected the morphology of the place' is not clearly understandable, and it may need rephrasing.

Line 280: 'relief' would more commonly read as 'height' or 'elevation'.

Lines 286-287: It is not clear whether 'The morphology of relief. . .' refers to the 'The morphology of Mount Petrano. . .' or more generally to 'Slope morphology. . .'.

Lines 300, 301, 352: Missing space before figure number.

Line 450: 'relatively to' would better read 'with reference to'.

Line 450: 'proposing ways' is not very clear.

Line 451: The meaning of 'small. . . trips' should be clarified. Of small length? Or duration? Or both? If so, I would suggest 'short' instead of 'small'.

Caption Fig. 3: I suggest to specify to which place or stretch of coast the photos refer; in addition, 'up' and 'down' should better read as 'above' and 'below'.

Caption Fig. 4: 'Panoramic' should better read as 'Panoramic view' (as in Fig. 2).

Caption Fig. 5: I suggest to add details to the caption, such as 'The proposed itinerary from. . . to. . .' or 'starting from and reaching. . .'.

Caption Fig 6: Readers who are not familiar with the area may hardly recognize the geographic elements mentioned in the caption, in particular the Bosso and Burano streams. I suggest either to rephrase the caption avoiding to make specific reference to the streams or to depict in the photograph the main geographic elements (like in Fig. 7).

Caption Fig. 7: a) 'Rossa Fm' should become 'Rossa Fm.'; b) 'the relief morphology that was formed by selective erosion' should better read 'the geomorphological features caused by selective erosion'.

Caption Fig. 10: 'the earthquake' should better read 'the earthquake of. . .' specifying the date of the event.

Caption Fig. 12: reference to Mount Vettore should be made in the caption.

Caption Fig. 14: details can be added, such as 'The fault plane at . . . side of Mount Vettore' or similar explanation.

Caption Fig. 15: details should be added about the location of the fractures at Mount Vettore.

---

## Referee Comment (RC2) · Paola Coratza (Referee) · 8 Jun 2020

General comments The paper illustrates a new approach in geoscience communication through artistic works and initiatives specially devoted to non-specialist in the Marche Region (Italy). The paper is worthy of interest, the topic is interesting and fits with the scope of the journal Geoscience Communication. The paper is well presented, English is correct, and the is a good balance between the various parts of the paper. Just some minor revisions would be needed to ameliorate the manuscript. Suggestions are reported below.

Specific comments Line 38: Even if the papers quoted are certainly of great interest,

[Figure]

the Authors could quote articles more updated: Please consider to quote the book "Geoheritage: Assessment, Protection, and Management" Reynard and Brilha (Eds), 2017 and chapters within (e.g. Chapter 5 - The Specificities of Geomorphological Heritage (Coratza and Hoblea); Chapter 8 - The Landscape and the Cultural Value of Geoheritage (Reynard and Giusti); Chapter 17 - Geoheritage and Geotourism (Newsome and Dowling). Line 29: The fullstop should be deleted before the parenthesis Line 71: The "ability" not seems to be the correct word. Line 78: "which are things" would be better read "variables such as..." Line 83: "The next step..." I suggest to go head with this sentence Line 90: "inherent aesthetic richness" would be better read "high inherent aesthetic value" Line 142: "S.Maria" add space Line 144: PU would be better read Pesaro – Urbino Province Line 162: "didactic" would be better read "educational" Line 176: delete full stop after Bartolo Line 275: "overwhelming; as the..." would be better read "overwhelming, as the..." Line 300: Missing space before figure number Line 301: Missing space before Mount and before figure number Line 352: Missing space before figure number Line 353: "Castel-santangelo" would be better read "Castel-Santamgelo" Line 369: "paleosurface" delete space

The following references are not quoted in the text: Line: 484: Bartolini & Pecerillo Line 525: Sala and Westley

---

## Author Comment (AC1) · 8 Jun 2020

Dear Mauro Soldati,

Thanks for your review and the very helpful suggestions. Your corrections have been much appreciated and all of them have been included in the new version of the paper.

In detail:

We modified, according with your suggestions, the Lines 142, 145, 165, 300, 301, 352, 450 and the captions of Fig. 4, Fig. 5, Fig. 7, Fig. 10, Fig. 12, Fig. 14, and Fig. 15;

Lines 166-167: The expression "aspects of landscape evolution that affected the morphology of the place" was simplified in "the landscape evolution";

Line 280: the term "relief" was replaced with "elevation";

Lines 286-287: "The morphology of relief..." was corrected with "The morphology of Mount Petrano";

Line 450: "proposing ways" was modified in "communicative methods";

Line 451: the term "small... trip" was replaced with "short", as you suggested;

Caption Fig. 3 was corrected in: "Figure 3: Two stretches of the San Bartolo cliff showing the pebbly beach in the northernmost sector (above) and the substrate eroded by the storm waves in southernmost one (below)";

Fig. 6: We identified in the photograph the main geographic elements (adding some letters) and we modified the figure caption in: "Figure 6: The Mount Catria-Mount Nerone anticline ridge, view from the southern side of Mount Nerone. A = Mount Petrano, B = Catria Massif".

---

## Referee Comment (RC3) · Sydney Lancaster (Referee) · 9 Jun 2020

This paper describes a project which uses music and poetry to increase interest in, and understanding of, the geological and geomorphological landscapes in the Marche Region (Italy). The project described employs a range of media and experimental means to convey information and appreciation of the landscape of the region and its geological processes to good effect. The paper and the project described therein are well suited for publication in Geoscience Communication.

There is mention made in this paper of the size and demographics of audiences attending public presentations of this work; the authors are encouraged to provide this data as a supplementary table if it is at all available. I look forward to reading about further developments in the next phase of this project, as the research will include more data from audience feedback and demographics; this information can be very useful in gauging the success of the project overall, and in determining the most effective elements of a complex project. I encourage the authors to gather and share data regarding online reception/viewing/comments for the project as well, as this can contribute to an understanding of the international reach of the work.

I would encourage greater detail in labelling figures and in the figure captions; for those of us unfamiliar with the locations, more information in both the figures and their captions would contribute to our understanding.

The manuscript is generally well organized and written, and requires only minor revisions - mostly to phrasing, and small technical corrections, which are suggested below. Please refer to the accompanying marked-up manuscript uploaded as a supplement to this review for recommendations regarding word choice and phrasing for clarity.

Technical Corrections line 38, add an 's' to reference line 80, add comma after equilibrium line 189, add comma after relief line 283, add comma after territory line 285, insert 'of' after type line 287, add comma after anticline line 288, add comma after ) line 397, insert 'and' before balance line 429, delete 'at the' and change mirror to mirrored line 437, replace period after strong with a colon; change It to it following colon line 438, insert 'the principle melody,' before concrete line 439, insert comma after movement line 448, insert comma after form

Please also note the supplement to this comment:
https://gc.copernicus.org/preprints/gc-2020-5/gc-2020-5-RC3-supplement.pdf

**Supplement:**

[revised manuscript text omitted]

---

## Referee Comment (RC4) · John Gordon (Referee) · 11 Jun 2020

This paper makes a valuable contribution in two respects: 1) in exploring new and engaging ways of promoting geoscience and geoconservation to the general public; and 2) in demonstrating links between geoheritage and cultural heritage.

1. My main comment is that the conceptual framework could be strengthened and more clearly set out in the introduction by reference to the wider literature on best practice in interpretation*, particularly in relation to stimulating emotional responses and encouraging memorable experiences involving a range of senses and interactive engagement. Emotional experience and making personal connections can be a powerful

basis for subsequent positive actions or behavioural changes by those participating - either people attending the events described or visitors to the sites. At present this is scattered through the text: e.g. lines 83/84 (establish a personal connection so that the public cares to further understand and appreciate the landscape); lines 445/446 (The communication of information of any nature through the emotional sphere is recognized to be much more effective than traditional communication methods); Lines 67, 73, 465 (foster desire to protect the landscape). *For example, building on Freeman Tilden's principles, a key reference here is: Ham, S.H. 2013. Interpretation: Making a Difference on Purpose. Fulcrum: Golden, CO, USA. And on the power of imaginative storytelling, see: Strauss, S. 1996. The Passionate Fact. Storytelling in Natural History & Cultural Interpretation. Fulcrum: Golden, CO, USA.

2. Evaluation of the methods adopted is critical in demonstrating the wider value of this type of approach and it is encouraging to see this noted as a next step (line 450).

3. I appreciate the focus of this particular paper is on promoting better awareness among local people, but I think the authors could draw more links with geotourism and point out the potentially wider relevance and value of their work to this field, particularly in the conclusion.

4. In places the text is written in a personal style (frequent use of 'you') and using emotive language (e.g. 'guzzling by the jaws of erosion'). I can see that this is appropriate for presentations to the public but is less suited for a scientific publication where more measured language is appropriate.

5. There are some further comments, edits and points of clarification which I have itemised separately to assist the authors in finalising the paper. I have also made suggestions for alternative wording for the authors to consider to help clarify the meaning.

Please also note the supplement to this comment:

https://gc.copernicus.org/preprints/gc-2020-5/gc-2020-5-RC4-supplement.pdf

**Supplement:**

Review: Science, Poetry, and Music for Landscapes of the Marche Region, Italy. Teaching the Conservation of Natural Heritage: Olivia Nesci and Laura Valentini

This paper makes a valuable contribution in two respects: 1) in exploring new and engaging ways of promoting geoscience and geoconservation to the general public; and 2) in demonstrating links between geoheritage and cultural heritage.

1. My main comment is that the conceptual framework could be strengthened and more clearly set out in the introduction by reference to the wider literature on best practice in interpretation*, particularly in relation to stimulating emotional responses and encouraging memorable experiences involving a range of senses and interactive engagement. Emotional experience and making personal connections can be a powerful basis for subsequent positive actions or behavioural changes by those participating - either people attending the events described or visitors to the sites. At present this is scattered through the text: e.g. lines 83/84 (establish a personal connection so that the public cares to further understand and appreciate the landscape); lines 445/446 (The communication of information of any nature through the emotional sphere is recognized to be much more effective than traditional communication methods); Lines 67, 73, 465 (foster desire to protect the landscape). *For example, building on Freeman Tilden's principles, a key reference here is: Ham, S.H. 2013. Interpretation: Making a Difference on Purpose. Fulcrum: Golden, CO, USA. And on the power of imaginative storytelling, see: Strauss, S. 1996. The Passionate Fact. Storytelling in Natural History & Cultural Interpretation. Fulcrum: Golden, CO, USA.

2. In the title, I suggest 'communicating' rather than 'teaching'. The approach of the study is on engaging and communicating with people rather than didactic methods.

3. Evaluation of the methods adopted is critical in demonstrating the wider value of this type of approach and it is encouraging to see this noted as a next step (line 450).

4. I appreciate the focus of this particular paper is on promoting better awareness among local people, but I think the authors could draw more links with geotourism and point out the potentially wider relevance and value of their work to this field, particularly in the conclusion.

5. In places the text is written in a personal style (frequent use of 'you') and using emotive language (e.g. 'guzzling by the jaws of erosion'). I can see that this is appropriate for presentations to the public but is less suited for a scientific publication where more measured language is appropriate.

6. There are some further comments, edits and points of clarification which I have itemised below to assist the authors in finalising the paper. I have also made suggestions for alternative wording for the authors to consider to help clarify the meaning.

Title
Suggest colon rather than full stop after Italy

Abstract
Line 10. arouses rather than arouse
Line 11. comma after Italy and which rather than that
Line 18. geosites rather than geo-sites; their rather than its
Line 19. arriving at rather than to; their rather than its; appreciation might be better than knowledge?

**1 Introduction**

Line 23. comma after audience

Line 23/24. might be better expressed as ...in several contexts: e.g. protests or movements promoting important social and environmental issues....

Line 27. Tracking used approaches might be better expressed as Evaluating existing approaches?

Line 29. raising people's awareness of complex topics; no full stop after topics

Line 35. no comma after time; learn about

Line 37. has expanded greatly rather than has been very numerous

Line 38. you could also refer to the publication by Reynard, E., Brilha, J. Eds. 2018. Geoheritage. Assessment, Protection, and Management. Elsevier: Amsterdam

Line 38. ideas have emerged

Line 39. no hyphen in geotourism

Line 41/42. Quotation should be in parentheses

Line 49. has produced; great might be better and more measured than amazing

Line 51. delete us

Line 54. suggest high rather than enormous biodiversity

Line 55. and geodiversity? I think you need to add geodiversity here

Lines 55/56. Maybe reword as ... was the birthplace of eminent historical and literary figures [if that applies] (e.g. name one or two), while many others (e.g. name one or two) have travelled through the area

Line 63. address rather than addresses

Line 65. Choices about what?

Line 66. dash rather than comma after communication; dash rather than comma after poetry

Line 67. people's desire?

**2 Objectives and methods**

Line 69. arriving at

Line 70. I don't understand what the problems and weaknesses of a place are? Do you mean its fragility, natural hazards, geological/scientific problems, socio-economic problems or what?

Line 71. Why must it? Some places are not necessarily attractive. Some may be awe-inspiring but not beautiful. Some discussion of landscape aesthetics and appeal might be appropriate here. The aesthetic appeal of physical features has been an important factor in tourism over the last few centuries (from the Romantic movement onwards) and more recently in geotourism

Line 75. insert and before the culture

Line 77. follows rather than proceeds through?; comma after view

Line 78. delete which things.

Line 79. delete a before simple

Line 82/83. Providing the system...... is not a sentence

Line 83. End this paragraph after environment. Start new paragraph as The second step is to establish... and run on to next sentence This second step

Line 87. suggest insert and before more

**2.1 The working method**

Line 90. Three sites or Twenty sites?

Line 92. in the region

Line 95. delete a

Line 97. Do you mean characteristics rather than peculiarities?

Line 98. key words relating to the place; the atmosphere it evokes

Line 100. delete the before musical

Line 102. Delete as hereafter described

Line 104. Insert is before aptly

Line 106. At other times; comma after times
Line 107 is rather than was; associations rather than the association
Line 110. Suggest foster love for a place rather than make you love a place
Line 111. emerges
Line 112. natural rather than naturalistic
Line 114. elements rather than contents?
Line 116. Delete we; delete of this work
Line 117. Delete just published
Line 118. Delete same
Line 119. which can be enjoyed rather than you can enjoy
Line 121. where the book is also available in interactive form
Line 122. results rather than contents? no comma after events
Line 123. the individual places?

**2.2 Description of the events**
Line 124. Consider 'Live events' as an alternative section heading
Line 127. means of simple and popular language; reading rather than acting of poems; performance rather than performing
Line 128. delete of ancient music (this was explained above); The project, as described above, includes
Line 129. took place
Line 130. delete one; employment rather than employ
Line 133 has rather than have
Line 134. delete will; represent the links between the science.....
Line 136. delete in
Line 137. delete comma after public
Lines 140/141. Delete text from 'some of them.......locations but'
Line 141. involving rather than searching; what is a suggestive place?
Line 144 Fortress; comma after Maniscalco"
Line 145. Insert 'the' before program
Line 146. also from elsewhere in Italy?
Line 147. after potential, insert and is

**2.3 Experience with the public**
Line 151. delete a
Line 154. it is rather than it's
Line 154/155. Suggest rephrasing ....to quantify these factors because they depend on several variables (e.g. the advertising before the event, the season and the weather, the beauty of the place and how difficult it is to reach)
Line 156. What is receptive capacity?
Line 158. delete just? the book was published in 2019
Line 160. Suggest rephrasing...We have the material to propose and deliver public events and collect the responses of the participants.
Line 161. presented rather than proposed?
Line 162 schools and museums
Line 164. the shows are
Line 165. comma after Region); cultural event rather than moment?

**3 The three case studies**
Line 168. Suggest the title of the section is simply 'Case studies'

Line 169. Suggest shortening to.. Among the many amazing landscapes of Italy we focus on three case studies from....

Line 174. 200 m high
Line 176. delete comma afterBartolo
Line 178/179. Do you mean the beaches are eroded by the sea only during the strongest storms, whereas the less protected rocky ridges are more exposed to wave erosion?
Line 183. human activity rather than man? natural balance?
Line 182. Is it not that the anthropic causes are superimposed on the natural ones?
Line 186. sediment load rather than solid flow? and again line 186
Line 189. insert they before corrupt;
Lines 189-191. Meaning of this sentence is not clear
Line 192. during the Holocene (the last 11,000 years)
Line 194. the position of the paleo-coast.... advanced by about 2 km
Line 197. an interpretation route? or geotrail?
Line 201. insert and after interpretation and delete the comma; I don't understand the phrase 'other than the critical issues' and suggest it is deleted
Line 205. 'guzzling by the jaws of erosion'. I can see the reason for using emotive language in the public events, but this is a scientific paper
Line 207. The visual (colour), olfactory (scent) and physical (sky and sea) balance is represented...
Line 208. looks? visions might be better?
Line 275. on the previous? suggest delete
Line 276. insert there are before several

Line 282. delete the before Mount Petrano
Line 283. of or across the whole Province?
Line 285. example of an anticlinal ridge; one type of folded rock layers; forces that act slowly over a very long time period
Line 288. comma after 7)
Line 289. the Maioloca Formation
Line 290, The rocks of the Marne a Fucoidi Formation, which rests on the Maiolica Formation,...
Line 291. The rocks above are again more resistant, comprising ......
Line 297. delete comma after Petrano
Line 299 the Apennines rather than our Apennines
Line 300. space after Fig.
Line 301. space after Fig.;  delete the before Mount Petrano
Line 303. suggest replace you with the visitor; From there the visitor can experience..
Line 304. delete the before stop 2; the visitor rather than you?
Line 306. characteristics rather than peculiarities?
Line 306. key words rather than keys?
Line 309. What does reminiscent one day holiday mean?
Line 339. Suggest reword as: The music selected was twelve variations.......KV 265, which is a keyboard.....
Line 359. 5 km; 2 km
Line 368. semicolon after depression; Middle Pliocene -Lower Pleistocene is ~3.3-0.7Ma. If referring to the time interval, use Early Pleistocene
Line 369. low-energy; paleo-surface
Line 370. what is a direct fault system?
Line 371. by intense uplift; disrupted rather than dismantled?
Line 375.oriented N-S
Line 377. Suggest give date for the Middle Pleistocene (~0.7 Ma)

Line 377. Suggest re-wording - filled with debris deposited by streams flowing off the adjacent slopes. The debris originated from glacial and periglacial processes as evidenced from.....
Line 380. at depth
Line 381. which rather than that
Line 383. the Sibilllini's thrust plane. What is this?; in all its majesty?
Line 387. for excursions or for visitors, rather than excursionists
Line 389. delete the before Mount Vettore, and again line 392
Line 393. the left
Lines 394/395. form open ground fractures 30-40 cm wide
Line 396. key words; characteristics rather than peculiarities?
Line 400. The natural cycle of nature tends to plan ..... The meaning of this sentence is unclear
Line 424. The music selected is Johann.....
Line 428/429. heights?
Line 433. delete here proposed

**4 Discussion and conclusions**
Line 441. seeks to promote
Line 443. You could make a stronger statement here - Art has great power......and provides an powerful means to communicate specific subjects.
Line 447. comma after background; indicates rather than uphold? landscape origins rather than problems posed?
Line 450. delete able; interpretation methods rather than proposing ways?
Line 456. described above
Line 458. to rather than inside

**References**
The references to Bartolini & Peccerillo and Sala & Westley are not cited in the text.
The references to Curtis and Curtis et al are not in alphabetical order.

**Figure captions**
**Figure 2: Panoramic view of Mount San Bartolo.**
**Figure 3: Pebbly beach (upper) and substrate eroded by waves (lower).**
**Figure 4: Panoramic view of the active cliffs of Mount San Bartolo.**
**Figure 6: The Apennine ridge seen from Mount Nerone. Mount Petrano is located between the incision of the Bosso and Burano streams. Mount Catria is in the background.**
**Figure 7 - delete bracket after Scaglia Rossa Fm**
**Figure 9: Delete the before Mount Petrano**
**Figure 13: Delete the before Mount Vettore**

---

## Author Comment (AC2) · 15 Jun 2020

Dear Paola Coratza,

Thanks for your review and the very helpful suggestions. Your corrections have been much appreciated and all of them have been included in the new version of the paper.

In detail:

We quoted the chapters of the book "Geoheritage: Assesment. . .", many thanks for the advisement. We modified, according with your suggestions, the Lines 29, 71, 78, 83, 90, 142, 162, 176, 275, 300, 301, 352, 353, 369;

[Figure]

Line 144: we deleted the reference to the Province, maintaining only that to the Marche Region; We removed the indicated references, which were not quoted in the text.

---

## Short Comment (SC1) · 29 Jun 2020

Dear Olivia Nesci and co-authors,

During a recent virtual writing retreat, we used a peer-review framework to review your abstract. We then had an open discussion and noted down all the feedback. We reviewed your abstract using a structured worksheet with the following advice in mind: "The abstract is a condensed and concentrated version of the full text of the research manuscript. It should be sufficiently representative of the paper if read as a stand-alone document". We looked for important elements of a research abstract and we comment on them below. We hope the following is helpful for your revisions. It's important to note that Geoscience Communication puts a lot of emphasis on evaluation of communication practice and ensuring that the practice is based on a solid research question and research design. The articles need to tell the story of research on geoscience communication and not just tell the story of geoscience communication that's been done.

Overall: Your project sounds very interesting and we were really interested in this "new approach" and how it combines these different media. The abstract touches on some very interesting elements and issues, which made us want to read on. However, there are a few things we believe should be improved for this to be relevant for a peer-reviewed publication.

Title: The title reflects the abstract rather well as it stands. However, we were left wondering how important the idea of "conservation of natural heritage" is, and why this is not explained (or used) in the Abstract below. We also failed to understand what the focus of the paper was (which we also reflect upon below). Is the paper about teaching methods or is it about creating a personal connection or appreciation for the geo-sites in question?

Need and relevance: We understood clearly that the project has produced lots of interesting outputs, but the Abstract does not tell the reader why. At the moment, the main relevance of this project seems to be linked to the larger project that you describe. We would suggest that you should move information about the larger project from the Abstract and to the main body of the paper. You should focus on the need and relevance of your project in particular. Why is this "new approach" needed and what might it achieve?

Question/Hypothesis The main question for the research behind this "new approach" was also unclear. We wondered if the main aim was to create a new pedagogical resource, but we could not be sure. The main aim (and the question behind it) needs to come across a lot clearer.

Methods: We assume that the "plain language accompanied by visual stimulations, poetry, and ancient music" are the main methods used. However, it would be nice to have an idea about why these were chosen and how they have been evaluated. The point about evaluation is particularly important when you mention that the methods "arouse an emotional and intellectual experience that enables a personal connection to the place". We basically want to know exactly how this "new approach" was developed and evaluated. Hopefully you can easily extract this information from the main bulk of the paper.

Results and conclusions You start by saying that you develop a "new approach in science communication". One of the main results should therefore be a description and evaluation of this "new approach". It would be great if the Abstract could contain some more information about this.

Take-home message We agreed that the last couple of sentences in the Abstract mirrored the same message as the first couple of sentences. In the end you say that you "intend to educate people to have a new perception of geo-sitesÂ". However, it would be nice to have a statement at the end, which states whether you actually succeeded in educating the people to have this new perception. In other words, we want to hear whether your "new approach" actually had the impact that you intended in the beginning.

Clarity We thought some sentences were quite repetitive (as we mentioned directly above). Clearing up these sentences would free up space for you to expand more on the need, relevance, aim/question, and methods behind this study. Some of us also saw some sentences which could be reduced in size. For example the sentence " The project is documented through live multidisciplinary performances, the publication of project materials in a book with a DVD attached, and through a web site with the same contentsÂ" could be edited to "The project resulted in live multidisciplinary performances, a book, a DVD and a web site."

We hope you find these comments and suggestions useful. We really liked the sound of the project from the Abstract! We really hope that you can clarify some of the important points that we have mentioned to ensure that the Abstract contains all the important elements that it should and that it reflects the paper in a just and fair way.

Good luck!

Please also note the supplement to this comment:
https://gc.copernicus.org/preprints/gc-2020-5/gc-2020-5-SC1-supplement.pdf

---

## Author Comment (AC3) · 3 Jul 2020

Dear Sydney Lancaster,

Thanks for your review and the very helpful suggestions.

Your corrections have been much appreciated and all of them have been included in the new version of the paper.

In detail:

We modified, according with your suggestions, the Lines 30, 37, 47, 80, 89-91, 98, 178, 186, 189, 285, 300, 381, 394, 397, 435, 437, 438, 439, 447;

[Figure]

Line 38: we modified the reference list;

Line 150: yes, it will be our priority;

Line 280: the term "relief" was replaced with "elevation";

Line 284: not the exposed outcrop but the entire morphostructure;

Line 287: the term "relief" was replaced with "Mount Petrano";

Line 446: No, we still don't have; at the moment it is only our perception by talking with the public;

Line 450: "proposing ways" was replaced with "communication methods";

Fig. 2: We added some information in figure caption. The scale bar could be misunderstood in photographs; some elements of the landscape provide this kind of information;

Fig. 3, 7, 8, 11, 14 and 15: ok, we provided more details in figure caption;

Fig. 4, 6 and 12: more details in the photo and in the figure captions are provided.

---

## Author Comment (AC4) · 20 Jul 2020

Dear John Gordon,

Thanks for your accurate review and the very helpful suggestions. Your indications have been much appreciated and almost all of them have been included in the new version of the paper, except few cases where the text was already been modified following the suggestions of the previous referees.

In detail:

Point 1: we took in consideration, as you suggested, some literature in references
(Ham, 2013 and Strauss, 1996) and we inserted some sentences linked to these works in the text (lines 96, 457, 175, 460), aimed to strength the problem to establish a personal connection with the public and to stimulate a further understanding and appreciating of the landscape;

Point 2: we modified the title as you suggested;

Point 3: we underlined in the text that the next step would be the collecting of data about the response of the public relating to our method.

Point 4: we drawn more links with geotourism in the text (Lines 24, 45, 47);

Point 5: we modified the text according to yours suggestions: we avoided the personal style (use of 'you') and the use of emotive language;

Point 6: we modified, according with your suggestions, the Lines 10, 11, 18, 19, 23, 24, 27, 29, 35, 37, 38, 39, 41-42, 49, 51, 54, 55-56, 63, 66, 67, 69, 70, 71, 75, 77, 78, 79, 82, 83, 87, 95, 97, 100, 102, 104, 106, 107, 110, 111, 112, 114, 116, 117, 118, 119, 121, 122, 123, 124, 127, 128, 129, 130, 133, 134, 136, 137, 141-141, 141, 144, 145, 146, 147, 151, 154, 155, 156, 158, 160, 161, 162, 164, 165, 168, 169, 174, 176, 182, 183, 189, 189-191, 192, 194, 197, 201, 207, 208, 275, 276, 282, 283, 285, 288, 289, 290, 291, 297, 299, 300, 301, 303, 304, 306, 309, 339, 359, 368, 369, 370, 371, 375, 377, 380, 383, 387, 389, 392, 393, 394-395, 396, 400, 428-429, 433, 441, 443, 447, 450, 456, 458;

Line 65: we deleted some words in the sentence;

Line 90: we modified the text, to avoid the misleading;

Line 98: the text was already been modified according to another referee;

We corrected the mistakes in the references list;

Some information in the figure captions and some new details in the photographs are provided; we modified the text following your suggestions.

---

## Author Comment (AC5) · 20 Jul 2020

We thank the team of reviewers who gave us good advice on the title and the abstract. We have considered their comments even if the latest version of the text has already been modified according to the suggestions of the four reviewers chosen by the editor and editor herself.

Title: the concept of heritage conservation was explained in the abstract. The term teaching is not strictly related to "school methods" but rather "to teach" ordinary people to appreciate geographic sites in a new way.

[Figure]

Abstract: we answered the question: (1) how this "new approach" was developed and evaluated and (2) Why poetry and ancient music were chosen and how they have been evaluated? (3) Did this "new approach" actually have the impact that you intended in the beginning?

(1) Our approach is based on the awareness that the arts can help synthesize and convey complex scientific information and create a celebratory and positive atmosphere. Evidence suggests that the arts can deeply engage people by focusing on the emotions rather than on the comprehension, which is often emphasized in science education;

(2) We believe that the choice of poetry and music as arts is motivated by contingent needs, in fact all the arts (visual: painting, architecture and literary: narrative and so on) could be similarly used. The partnership that gave birth to this new approach is made up of people who practice music and poetry;

(3) The social impact that this new approach expected at the beginning of this research has been remarkable and confirm its effectiveness. The high participation and sharing in social networks and the attendance by a very large and varied audience, mostly without a scientific background at our live shows, demonstrated a great interest in the problems proposed.

---

## Author Response (AR1)

Dear Tiziana Lanza,

Thanks for your useful review and very interesting suggestions. Your indications have been really appreciated, and included in the new version of the paper.

In detail:

**Abstract**

We followed all your suggestions. We deleted part of the affirmation "The high participation and sharing in social networks and the attendance by a very large and varied audience, mostly without a scientific background, at our live shows, demonstrated a great interest in the geological history, resulting relevant for the development of geo-tourism." We agree that, since we still did not collect the feedback of participants in a systematic way, this phrase could convey the readers to expect numbers, graphics and tables in the following text.

We added some words to stress the importance of the conservation of the natural heritage (as Reeve suggested).

Answering to Reeve: the abstract explains that the paper concerns about a teaching method and we illustrate some case studies, where we create a personal connection for the geo-sites in question.

We inserted not here, but in the text, some data coming from the YouTube channel and from social network.

**Objectives and methods**

We modified the title in "Motivation and Objectives", and changed accordingly the contents of the paragraph.

We insert some sentences trying to answer to the requests of Reeve in "need and relevance" (Why is this "new approach" needed and what might it achieve?). We avoided speaking of a "new" approach, but we stressed some aspects of the working method of "our" approach. We also tried to clarify the main aim of our work.

A section of the paragraph (Our work follows two different routes. The first step analyses the landscape from the scientific point of view…………………..  making scientific information without complexity and associating it with the interpretation that poets and musicians have given.) was replaced in the following paragraph: "The working method".

**The working method**

We tried to explain better the method. For instance, we add information about our choices: why we selected among the arts the poetry and the ancient music ? Why we always start the work on the sites from the geology? Why we propose original poetry and not-original music?

We followed the suggestions of Reeve trying to clarify how our approach gave a positive response. We tried to avoid repetitions in the text, while extending the information about the group. We added some data on the timeline, underlining that our project is still young and, at this step, we do not have enough information to give numbers on the feedback of the public.

**Experience with the public**

We insert some sentences explaining why the feedback could be important in the next step, and how we intend to collect data on the feedback.

We add a photo acquired during one of the live shows.

[revised manuscript text omitted]

---

## Editor Decision (ED1)

I would like to thank all the reviewers for their interesting and useful suggestions. I concentrate mainly in the last three reviews, since they address the issue of the evaluation of the method.

Lancaster writes: "there is mention made in this paper of the size and demographics of audiences attending public presentations of this work; the authors are encouraged to provide this data as a supplementary table if it is at all available".

I invite authors to meet the request if they are able to.

Gordon encourages to strengthen the conceptual framework by reference to the wider literature on best practice in interpretation, particularly in relation to stimulating emotional responses and encouraging memorable experiences involving a range of senses and interactive engagement. I believe that the authors have accepted this suggestion. But I would like to stress what Gordon says: "Emotional experience and making personal connections can be a powerful basis for subsequent positive actions or behavioural changes by those participating - either people attending the events described or visitors to the sites". This assertion is corroborated by studies as Gordon himself underline in his comment.

To meet Reeve claims: "The point about evaluation is particularly important when you mention that the methods "arouse an emotional and intellectual experience that enables a personal connection to the place". We basically want to know exactly how this "new approach" was developed and evaluated.

In evaluating the method, communication using art can already count on the powerful basis already underlined by Gordon. I would like also to remind that this is a paper for a Special Issue that particularly address works on Earth sciences and Art, a field that is "work in progress", and that eventually can bring to the contamination of the two methodologies in approaching the public. In invite to read my introduction to the volume and the studies I quoted already performed on the synergies of such collaboration. Moreover, most of the projects have been carried out by artists in collaboration with scientists. It is not mandatory that scientists are also science communicators, while I believe that for an artist to make a study on the impact of his work on the public it is not fundamental. They are not accustomed to do an evaluation of the social impact of their artworks, at least from a scientific perspective.

Nevertheless, to encourage authors to meet at their best the issues discussed and concerning the evaluation of the method and the need and relevance of their work, I suggested directly to them further review.

---

## Author Response (AR2)

Dear Sam Illingworth,

Thank you very much for your useful suggestions. The paper was modified accordingly, in particular:

Line 36: we inserted "of";

Lines 42-43: the text was modified as you suggested;

Line 76: "place" was changed in "landscape";

Lines 96-104: the text was modified according to your suggestions; one sentence was modified and moved in Line 102;

Line 110-115: we followed your suggestions;

Lines 148-149: it means that the shows are dedicated only to five-six places, for reason of time. We clarified;

Lines 170-173: we modified the text following your suggestions; we provided evidences for the number of views;

Line 182: "to make" was changed in "in making";

Line 188: the sentence was modified as you suggested;

Line 193: the sentence was clarified;

Lines 194-195: the sentence was changed as you suggested;

Lines 471-478: we modified the text following all your suggestions;

Line 483: we clarified "study region" changing with Marche region;

Lines 490-493: the sentence was rewritten as you suggested.

[revised manuscript text omitted]

---

## Editor Decision (ED2)

[revised manuscript text omitted]

Line 149 – 150

please clarify this concept. Does this mean that the shows are dedicated only to five-six places, or do you mean that up to now only events linked to these particular places took place?

Line 171-174

Our experience, therefore, is based on direct feedback from the public. They always show a genuine interest asking questions and further info by email. Social media are also an indicator of a widespread interest in our project, with 27,000 views on the whole. While our Facebook page has 871 followers.  (can you provide evidence of the 27,000 views?)

Please clarify line 193

Line 473

Our experience confirms this effectiveness.  The presence at our live shows of a very large and varied audience, mostly without a scientific background, indicates a great interest in the origins of the landscape.

Line 483 please clarify "study region"

People of all ages and background can participate in this experience to gain awareness of the cultural heritage that the landscape represents. The ultimate goal is to increase their desire to understand the fragility of the territory, stimulating, at the same time, the will to protect and preserve it.

[Figure]

625

**Figure 16: Live show in the theatre of the prominent Renaissance Fortress of Sassocorvaro (PU).**